# Conceptual Framework for Future WSN-MAC Protocol to Achieve Energy Consumption Enhancement

**DOI:** 10.3390/s22062129

**Published:** 2022-03-09

**Authors:** Abdulrahman Sameer Sadeq, Rosilah Hassan, Hasimi Sallehudin, Azana Hafizah Mohd Aman, Anwar Hassan Ibrahim

**Affiliations:** 1Center for Cyber Security, Faculty of Information Science and Technology, Universiti Kebangsaan Malaysia, Bangi 43600, Malaysia; p94322@siswa.ukm.edu.my (A.S.S.); hasimi@ukm.edu.my (H.S.); azana@ukm.edu.my (A.H.M.A.); 2Department of Electrical Engineering, College of Engineering, Qassim University, Buraidah 51411, Saudi Arabia; dr.anwar@qec.edu.sa

**Keywords:** MAC protocols, Wireless Sensor Networks, performance, energy harvesting, IoT, network performance

## Abstract

Nowadays, the rapid deployment of Wireless Sensor Networks (WSNs) and the integration of Internet of Things (IoT) technology has enabled their application to grow in various industrial fields in our country. Various factors influence the success of WSN development, particularly improvements in Medium Access Control (MAC) protocols, for which WSNs-IoT are deemed vital. Several aspects should be considered, such as energy consumption reduction, performance, scalability for a large deployment of nodes, and clustering intelligence. However, many protocols address this aspect in a constrained view of handling the medium access. This work presents a state-of-the-art review of recently proposed WSN MAC protocols. Different methods and approaches are proposed to enhance the main performance factors. Various performance issue factors are considered to be the main attribute that the MAC protocol should support. A comparison table is given to provide further details about using these approaches and algorithms to improve performance issues, such as network throughput, end-to-end delay, and packet drop, translated into energy consumption.

## 1. Introduction

The continuous enhancement of smart sensors and low-power electronics has confirmed Wireless Sensor Networks (WSNs) position as one of the top-ranked technologies in the last few years [1]. These sensors are tiny, low-cost, low-power, multi-functional devices with short-range communication capabilities [2]. The impressive features of WSNs have made them a fundamental part of smart networks, where they can be used in automated testing, data processing, and small wireless transmission units in senor devices [3]. WSNs are composed of a vast number of sensors that can sense the environment and collect data, store them, and send them for further processing [4]. Data collected by the sensor devices are routed toward the sink node in a hop by hop approach. The sink node, which is the base station, performs further processing and analysis [5].

Applications that use sensor devices have evolved significantly in various fields, including healthcare surveillance, military surveillance systems, home automation, industrial systems, and environment monitoring systems [6]. Nowadays, researchers are working more on new implementations in different WSN application areas [7], as shown in Figure 1.

Transferred data in WSN networks are heterogeneous; for example, sensor nodes can be used to capture and transfer video, images, and audio for various applications [8] and for surveillance and monitoring purposes. Sensor devices can be deployed inside machines to monitor machines’ statuses in factories or industries and automate the manufacturing process. Sensor devices have been used underwater to facilitate navigation, surveillance, and pollution monitoring and prevent disasters [9]. Other implementations of WSN include mobility. Some WSN nodes are mobile and do not have a fixed location; this kind is referred to as Mobile WSN [10].

The general characteristics of such application domains are that they have various constraints in terms of nature and requirements, requiring a specific level of quality in terms of energy efficiency, data delivery delay, reliability, bandwidth utilization, scalability, adaptivity, and throughput. Securing Quality of Service (QoS) is one of the critical challenges in resource-limited networks; thus, extensive studies have been proposed to enhance the QoS parameters and maintain the accepted level of service by researchers in recent years [11]. The enhancement of the Medium Access Control (MAC) layer occupied a major part among them, as it provides shared medium access, and all other upper layers, networks, and transport are limited to the MAC layer to provide secure QoS requirements [12]. Therefore, this review explores the recently proposed MAC layer enhancement to guarantee QoS parameters in WSN networks. WSN depends on a shared medium to send data which is managed by the MAC layer of the OSI (Open Systems Interconnection) layers model, where CSMA/CA (Carrier Sense Multiple Access/Collision Detection) algorithms are implemented to minimize the data loss by avoiding collisions. In WSN networks, the largest portion of the energy is consumed in transmission [13]. Therefore, critical QoS parameters such as delay, throughput, packet delivery ratio (PDR), and energy consumption mainly depend on the MAC layer’s design, which means that the MAC is considered the fundamental part of WSN application design [14].

Various review papers have been attempted to survey the existing MAC protocols design; however, a comprehensive review of MAC protocols dedicated to WSN which mainly focuses on different approaches to handling performance issues in WSN is still missing in recent research. Continuous improvements and new approaches have not been deeply investigated in the literature. This paper investigates state-of-the-art approaches proposed to enhance the performance of the MAC protocol in WSN. Various survey papers have been published for different purposes. Huang et al. [15] provided a summary of some recently developed MAC protocol designs for WSN. On the other hand, [16] focused on energy efficiency in the design of the MAC protocol. Meanwhile, [17] summarized the WSN MAC protocol and categorized it based on the problems needing to be solved. Next, the authors of [18] presented the latest advances in the MAC protocol for energy harvesting-based WSN.

The rest of this paper is organized as follows: WSN MAC protocol is presented in Section 2, the optimization of the MAC protocols is described in Section 3, research gaps and limitations are discussed in Section 4, and Section 5 concludes the paper and discusses the future scope of our work.

## 2. WSN MAC Protocol

The MAC protocol refers to a method used to allocate the use of a medium between the devices of a network; it performs a function similar to that of a chairperson of a meeting, whose responsibility is to recognize each speaker in turn and assure that only one speaker is talking at a time. In networking, MAC is crucial for organizing a device’s access during the allocation to the medium and preventing collision, and its importance increases as the number of devices increases [16]. TDMA (Time Division Multiple Access) and CSMA (Carrier Sense Multiple Access) are two main types of MAC protocols. TMDA mainly depends on a time-triggered paradigm, so it needs very accurate duty cycling and the coordinator needs to broadcast a beacon packet periodically, even when the sensors have no data to transmit. On the other hand, CSMA depends on an event-triggered paradigm, where a lower burden is placed on the communication, meaning that it achieves a lower collision rate and hence a higher performance, especially in a loaded network with many nodes.

The MAC protocol’s design mainly depends on the application type, such as traditional wired computer networks, with QoS parameter efficiency and network throughput being priorities in the design. The efficiency of energy consumption and network lifetime are major concerns in battery-powered WSNs [19]. Meanwhile, load balance and energy neutrality are considered major issues in energy harvesting WSN [20]. MAC protocols are categorized into several categories, as shown in Figure 2.

MAC protocols are divided into two main subcategories: synchronous MAC and asynchronous MAC. It depends on the synchronization type used between diverse nodes in the network. Different widely used synchronization schedules among network hosts are not widely deployed, because for energy harvesting the rate and availability mainly depend on the uncontrolled external atmosphere [21]. Therefore, the most appropriate protocol is the asynchronous protocol. Asynchronous protocols are classified into two parts: sender-initiated and receiver-initiated. Receiver-initiated protocols have numerous advantages compared to sender-initiated ones. As idle listening is low in this situation, the time it takes for the nodes to use a channel is much less than that for sender-initiated protocols [22]. As a result, more nodes can communicate with one another, increasing the network’s capacity and throughput. Due to the receiver control access, it is easier to identify collisions and retrieve data that have been lost. There is also less overhearing because the flare is shorter than the preamble. Several critical parameters are essential for performance when designing an MAC layer protocol.

Throughput: Throughput is a rate measure in which data or packets are sent to measure the efficiency of a protocol. It can be associated with capacity in wireless link deployment. The throughput should be at the maximum level for an efficient protocol [23].Scalability: This refers to adapting a protocol to results of increasing network size. Increases in traffic, overhead, and load are among the consequences of increasing a network’s size. Therefore, localizing interactions so that nodes require less global knowledge to operate is one way to address this issue [24].Latency: The time difference between message arrival and message transmission is referred to as latency. This is a crucial limitation for time-critical applications. In the case of a real-time environment, latency should be carefully determined and minimized.Hop count: The hops number taken by a packet to reach a sink is one of the constraints faced. The operation of the MAC protocol for single-hop and multi-hop scenarios is different. In the case of multiple hops taken to reach a sink, the data need to be aggregated before sending them to the sink [25].Load balancing: In multi-hop networks, proper traffic load distribution is critical, since an unequal load might lead to the complete depletion of energy. The load should spread evenly depending on the residual energy load.Error detection and correction: It is another duty of the MAC layer in conventional computer networks to test the correctness of the information received; however, in WSN, MAC’s predominant focus is achieving the energy utilization only [26].Packet loss: This occurs when one or multiple transmitted packets fail to reach their intended destination, where the packet delivery ratio is reduced to the bare minimum.Energy harvesting: This refers to converting energy into electrical energy to power electronic devices or circuits of the devices. If the surrounding sources are insufficiently harvesting energy to balance the energy consumption, those nodes will run out of energy, resulting in node death.

## 3. State-of-the-Art Approaches for MAC-Protocols Optimization

The literature has concentrated on the optimization of MAC protocols in the networks of wireless sensors for critical applications in areas such as emergency alerting and monitoring as well as health care [27,28]. Those subcategories are general approaches, game theory-based approaches, heuristic-based approaches, meta-heuristic-based approaches, machine learning-based approaches, and MAC scheduling in cross-layer approaches.

### 3.1. General Approaches

One of the earliest examples of energy-efficient MAC protocols is Sensor-MAC (S-MAC) [29], which aims to lever the long inactivation period of sensor nodes and their sudden activation when an event occurs and they need to send data. S-MAC uses three techniques to reduce the energy consumption of nodes. Firstly, it enables the periodic sleeping of nodes based on auto-synchronizing sleep schedules. Secondly, it uses in-channel signaling to allow each node to sleep when its neighbors transmit to another node. Thirdly, it applies message passing to reduce contention latency. Another improvement of the S-MAC protocol is converting its duty cycle to adaptive based on the non-occurrence of activation for a time threshold TA (threshold algorithm) [30]. TA is a remarkably simple algorithm that is optimal in a much stronger sense.

In the context of algorithmic and protocol design, the work of [30] considered the limitations of the S-MAC protocol [31], which include its considerable overhead due to the synchronization packets used. As a result, two enhancements were introduced to transform the protocol from S-MAC to ADMC-MAC, in which the duty cycle was set to have a run time based on the residual energy and the node’s queue length as input parameters. In addition, a node with a large number of packets in the queue and the most energy among the virtual cluster’s nearby nodes was chosen as cluster head. The results showed a lower performance in terms of packet delivery ratio compared to the original S-MAC. Another improvement of S-MAC is represented by Timeout-MAC (T-MAC) [32], which enables adaptive active duty cycles based on listening for time period TA and allows the system to sleep if no event has occurred. This improvement falls under the category of approaches that enhance MAC energy consumption based on monitoring traffic conditions and buffering and responding accordingly. However, such a category is criticized for causing latency, making it not suitable for many applications that are of a real-time nature.

In [33], the energy-efficient MAC protocol was accomplished by enabling a scheduling table for the node’s sleep/wake-up time and dividing the channel into a set of TDMA slots where each slot is provided for contention sub-nodes using CSMA/CA. Similar to S-MAC, this protocol suffers from overhead due to the need to update the scheduling table. An earlier work focused on improving MAC is Zero Collision-MAC (ZC-MAC) [34], which aims to achieve zero collisions based on decomposing the medium to a pre-defined number of slots with a size equal to the number of nodes by remembering the slots that collided in the previous cycles. Such an approach may result in failure in the case where there are a variable number of nodes and the number increases to become higher than the number of slots.

In addition, synchronization between the nodes is necessary with respect to the start/end of the virtual slots. A more recent variant is the approach designated learning-MAC [35], which aims to learn the probability of selecting slots for transmission based on success and collision. This approach utilizes a formula that enables a successful slot to be selected consistently as long as it is successful; otherwise, it decreases the probability of selection in a geometrical factor. A more recent work in this area is the study of [36], who modified CSMA/CA to enable deterministic back-off after successful transmission. This approach is designated as CSMA/ECA. This method involved integration with hysteresis and fair share by instructing nodes to send a certain number of packets according to a schedule to enable fairness.

An energy-efficient contention-based ADMC-MAC protocol was presented by [31]. This protocol is easier to adapt to mission-critical applications, since it improves data transmission throughput based on traffic conditions and node queue size and improves energy efficiency. This protocol corrects a flaw in the standard MAC protocol. In this protocol, two algorithms are proposed. The first approach is priority-based, involving picking a node as the cluster’s head that has a large number of packets in the queue and the most energy among the virtual cluster’s neighboring nodes. The second algorithm employs a regression technique to estimate a node’s duty cycle based on the traffic conditions and residual energy. Only a few nodes are involved in the packet transfer. As a result, the number of overloaded packets at intermediate nodes is reduced.

In [37], the authors present a centralized mechanism to schedule Time Slotted Channel Hopping (TSCH) time-slots with the optimal usage of resources. The suggested approach relies on full sub-graphs formed from the topology’s collision matrix. Sub-links graphs can be scheduled in the same time-slot but on distinct channels at the same time. The suggested technique is performed using the Personal Area Network Coordinator (PANC), which is supposed to know the topology completely. The TSCH MAC has also been used to develop a Markov model for estimating transmission time and energy usage during frame transmission. The suggested method outperforms previous analogous systems in terms of performance.

The beacon slot collision problem was proposed in the work [38] based on presenting a non-conflicting beacon scheduling system based on association order. Distributed multi-channel Deterministic and Synchronous Multichannel Extension—Guaranteed Time Slots (DSME-GTSs) scheduling was also developed in this work; this allocates DSME-GTSs across multiple channels optimally. The objective is to minimize the number of time-slots used while maximizing the usage of available channels. The performance of the suggested mechanisms is investigated in terms of energy efficiency, transmission overhead, scheduling efficiency, throughput, and latency through simulations and is proven to outperform previous systems.

Table 1 lists a summary of the general approaches proposed for enhancing the performance of the MAC protocol when used for WSN.

### 3.2. Game Theory-Based Approaches

Game theory has been used for the goal of MAC scheduling for nearly a decade. For example, [39] proposed the simplified game-theoretic MAC (G-MAC) protocol for tuning the contention window based on the game theory model assigned to each node, with various actions, such as transmitting, listening, and sleeping, representing player 1. On the other side, player 2, which represents the remaining n-1 nodes, can select one of three actions: transmitting (which can be divided into successful or failed), listening, or sleeping. The goal is to enable a cooperative game by considering the energy as a utility function for minimization. The authors of this work provided an analytical model that can be used to obtain the optimal sleep probability equilibrium to guarantee convergence.

The game theory energy-efficient MAC protocol, which was named energy-efficient TDMA (G-ETDMA) based on game theory for intra-cluster WSNs, was proposed in [40] and mainly depends on a game theory approach to reduce node energy consumption. On the other hand, a study by [41] found that the sender node did not send any data until the receiver’s beacon signal was received. Therefore, data are routed on multiple paths in the data collection network because each sender receives a beacon signal from several nodes. An optimization framework is proposed to reduce energy waste for the most power-hungry network nodes in this context. Studies conducted by [42] allow adjustable system parameters to be set to achieve a fair equilibrium point that minimizes system latency and energy consumption by two times. As an illustration, this formulation was applied to six recent wireless sensor network (WSN) MAC protocols. Research by [43] proposed the use of multilayer nodes with distributed MAC, which is referred to as Distributed-MAC (DS-MAC), where the listening time of the nodes is controlled based on the neighboring communication. Game theory optimization helps address route loss constraints while selecting a route towards the base stations (BS).

Table 2 lists a summary of game theory-based approaches that have been proposed for enhancing the performance of the MAC protocol for WSN.

### 3.3. Heuristic-Based Approaches

In the work of [44], the authors integrated WSN scheduling with deployment. After using a meta-heuristic algorithm: Ant Colony Optimization (ACO), Particle Swarm Optimization (PSO), and a heuristic algorithm for deployment that considers the degree of coverage, they have proposed a heuristic scheduling algorithm based on grouping the sensors and considering the residual energy in each sensor.

In the work of [45], a multi-channel scheduling algorithm was proposed. For this purpose, the authors formulated Integer Linear Programming (ILP) optimization. In addition, they proposed heuristic algorithm based on Langford subset generation. The algorithm was attributed to a contention-free, multi-RF channel, and was heuristic. Their heuristic algorithm was claimed to give a lower execution time than ILP with a comparable performance.

In addition, the authors used CSMA/CA in a typical way without any model in order to change the back-timer. In the work of [46], heuristic-based scheduling using a greedy approach was proposed with consideration of the aspect of energy harvesting. More specifically, the network was considered to contain nodes with energy harvesting capability; however, the batteries were imperfect, leaking was possible, and nodes were heterogeneous. This work was pioneering in the networks of the MAC scheduling of harvest use and store (HUS). However, the optimization solutions provided were subject to local minima because of their greedy nature.

In the work of [47], a mathematical model for solving the converge cast problem in WSN was presented. The model considers the case application of target monitoring and aims to minimize scheduling time-slots. It is solved using pricing problems in a round-robin fashion due to the NP-hard nature of the optimization. The authors of this work reported two issues: real time concern and a lack of physical layer issue consideration such as fading and shadowing.

The WSN scheduling approach that depends on the network virtualization concept is proposed in [48]. It separates networks into profiles, with each profile indicating a group of nodes with a similar channel demand nature or characteristics. To put it another way, two types of profiles have been proposed: periodic and bursts. The mainframe was then separated into a contention-free frame and a contention access frame, each carrying a set of guaranteed time-slots, according to IEEE802.15.4. Then, to enhance each profile’s utility, an optimization process was carried out. To complete the optimization, the method takes a greedy approach.

Table 3 summarizes heuristics-based approaches proposed to enhance the performance of the MAC protocol for WSN.

### 3.4. Meta-Heuristic Based Approaches

Some researchers have aimed to optimize the performance of the WSN MAC layer using the concept of meta-heuristic searching. In meta-heuristic searching, the algorithm generates a set of random solutions, which we call different names according to the metaphor of the meta-heuristic searching algorithm—i.e., the population in a genetic algorithm, the swarm in particle swarm optimization, and the team in the league championship algorithm, etc. Each of the solutions in the set of solutions was evaluated based on the fitness function that needed to be optimized. Next, a selection process was carried out to select the elites, referring to the best solutions among the candidate solutions. The elites were used to generate a new set of solutions. The algorithm was iterated until convergence happened.

The definition of convergence is when no extra improvement in the fitness values of the best solutions is added. The differences between the followed approaches of MAC optimization in meta-heuristic searching depend on various aspects: the design of the solution; the formulation of the objective function; and the nature of the optimization—e.g., online vs. offline. In the work of [49], the design of the solution includes two types of information: the first one is the number of nodes in each group and the second one is the packet lifetime for each group. On the other hand, objective function formulation includes the QoS, which combines two terms: the Packet Deliver Ratio (PDR) and the end to end (E2E) delay. The nature of optimization is online optimization.

Another example is the work of [50], where the solution is designed for encoding which channel and time-slot are allocated to a certain node. The objective function is formulated as the E2E delay. The algorithm is carried out in an offline way before operating the network. Some researchers have used PSO for MAC scheduling based on TMDA. For instance, [51] used PSO to optimize the membership functions of the fuzzy system responsible for minimizing the energy consumption in IEEE 802.15.4. The trial-and-error process can criticize this work for finding out the best rule base of the system. Other researchers have proposed hybrid approaches for WSN scheduling based on meta-heuristic searching. We give an example of the work of [52], where particle swarm optimization was integrated with an evolution algorithm to minimize the delay. However, this approach was based on a centralized aspect with respect to the coordinator, which causes an overhead and more energy consumption to minimize the delay.

Table 4 presents a summary of meta-heuristics-based approaches proposed to enhance the performance of the MAC protocol for WSN.

### 3.5. Machine Learning-Based Approaches

The literature contains numerous approaches for MAC scheduling based on exploiting the power of the neural network. In the work of [53], a recurrent neural network was proposed to model the channel occupation and predict the best time for transmission. The authors used software-defined radio to monitor WiFi- channel and perform the training. This work can be criticized based on the significant stochastic component of the channel, which is modeled by the neural network inefficiently and lacks convergence. The usage of neural networks for scheduling has been studied in terms of 5G networks for scheduling multi-cell channel accessibility and minimizing user interference.

In the work of [54], an allocation was carried out for channel resources among multiple small cells to enable scheduling an uplink or downlink for users by each cell in a time-slot. In most NN-based scheduling, neural networks were used to approximate the resource evolution and predict the moment and configuration of transmission for the node. For instance, in the work of [55], frequency and time-slot resources are allocated for sensors based on an NN trained to predict the best channel–slot pair for transmission—i.e., the one that is associated with the minimum resource usage. In [56], cognitive radio technology and machine learning approaches are discussed, emphasizing their roles in increasing the wireless communication network’s spectrum and energy efficiency. Vanitha and Balakrishnan [57] proposed a hybrid medium access protocol (MAC) for gathering data in a UAV-based wireless network on overcoming the transmission time constraints between nodes by collecting and processing data from each node and acting accordingly.

Table 5 presents a summary of machine learning-based approaches that have been proposed for enhancing the performance of the MAC protocol for WSN.

### 3.6. MAC Scheduling in Cross Layer Design

The concept of Cross Layer Design (CLD) was proposed in the research of WSNs compared with classical networks. It refers to modifying the traditional OSI standard of layers of networks and customization according to the application’s need. As we can observe in Figure 3, there is intercommunication between layers in a cross-layer design, making it more efficient in fulfilling the application requirements than a classical layered network.

Figure 3 shows a conceptual design of a cross-layer network. Many researchers have had proposals for CLD that involve changing the layers and the architecture from one to another to support the application requirements. In some systems, these might be latency constraints, throughput, or emergency limitations. The survey of [58] provides a thorough discussion of CLD and the various related challenges.

Nosheen et al. [59] proposed a cross-layer design for a multi-hop, self-healing, and self-forming tactical network that minimizes the time of call setup, delivering collision-free communication and reusing the Time Division Multiple Access (TDMA) protocol empty slots. However, this can result in high delay and low throughput. Triwinarko et al. [60] proposed a cross-layer MAC design with transmit antenna selection (TAS) and transmit power adaptation (TPA). As an implementation of vehicle-to-vehicle (V2V) communication, we are considering spatial multiplexing zero-forcing Bell-labs layered space–time (ZF-VBLAST) via a Multiple-Input and Multiple-Output (MIMO) time-varying flat-fading channel. The authors of [61] propose a novel Full Duplex MAC protocol named priority-based multiple access (PBMA) which is based on prioritized messaging between different nodes.

Researchers have stated that there are various challenges in CLD; among them is the multi-hop communication in WSN, which makes it difficult to create a simple CLD design. In the work of [62], three types of developments are made: the first one is optimizing the transmission power of nodes, the second one is selecting relay nodes, and the third one is synthesizing the cross-layer method. This approach is applied in Wireless Body Area Network (WBAN). In the work of [63], a hybrid fuzzy and ant colony optimization method is proposed.

Cluster head selection is based on three criteria: residual energy, number of neighboring nodes, and link communication quality. The concept of clustering uses the criteria to reduce the number of nodes in closer clusters to avoid hotspots. This makes the protocol more energy-efficient. For the routing layer, the algorithm uses four criteria: distance from the current cluster head and main station, residual energy, the length of the queue, and the reliability of delivery. In this study, the MAC scheduling relies on information provided from the routing layer, making the design a CLD design.

Table 6 presents a summary of cross-layer-based approaches that have been proposed to enhance the performance of the MAC protocol for WSN.

## 4. Research Gaps and Limitations

To develop the framework of Wireless Sensor Networks, a cross-layer design in Medium Access Control (MAC) improvement approaches, this review thoroughly analyzed the state-of-the-art techniques used for identifying research gaps in the performance of WSN MAC protocols in recent papers by means of a grounded theory approach. Based on the available information, we subsequently propose a framework for identifying new trends and contributions of WSN-MAC to qualitative literature reviews and validate its application with an example. The outcomes of the review results identified the lack of research in following state-of-the-art approaches proposed to enhance the performance of the MAC protocol in WSN, thus enabling researchers to review current studies more rigorously and allow them to be effectively used in the future.

This paper mainly addresses the performance metrics and the enhancement process used in each approach. However, other issues need to be investigated in terms of radio block, data transmission, battery size, and microprocessor behavior. On the other hand, multiplayer approaches still face various challenges in terms of the coherence between individual layer standards for integrating with other layers. The validity of each approach in terms of its implementation against different deployments of WSN needs to be investigated further, as the specific approach cannot fit other application requirements. The overhead of some approaches needs to be investigated in terms of other performance metrics, such as computation power and energy consumption.

## 5. Conclusions and Future Work

Critical applications that depend on Wireless Sensor Networks need a high number of sensor nodes when deployed in existing environments. As these nodes have limited resources and access, performance is considered to be one of the most critical issues, as the node’s energy is an important aspect. Energy harvesting technology is a promising approach to this problem, as it supplies energy continuously to nodes over time. MAC protocol design, which must meet the requirements of networks, mainly depends on the application type; thus, design parameters must be carefully considered.

This paper provides a current overview of all WSN-MAC protocols. The factors behind various performance issues are the main attributes that MAC protocols need to support. A comparison table is given to provide further details about using various approaches and algorithms to mitigate performance issues, such as issues with network throughput, end-to-end delay, and packet drop, which are related to energy consumption. A comparative summary of the various MAC protocols and key aspects is presented in a categorized format in the hope that it will provide clear guidelines for interested researchers. Future research should aim to examine these parameters closely and investigate how these operations can be enhanced.

## Figures and Tables

**Figure 1 sensors-22-02129-f001:**
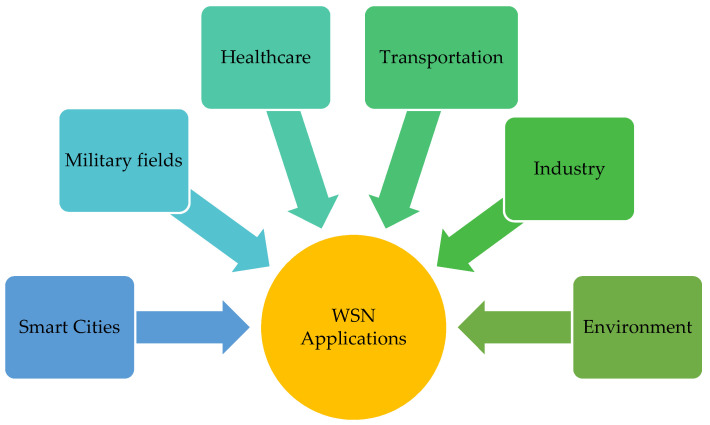
WSN application areas.

**Figure 2 sensors-22-02129-f002:**
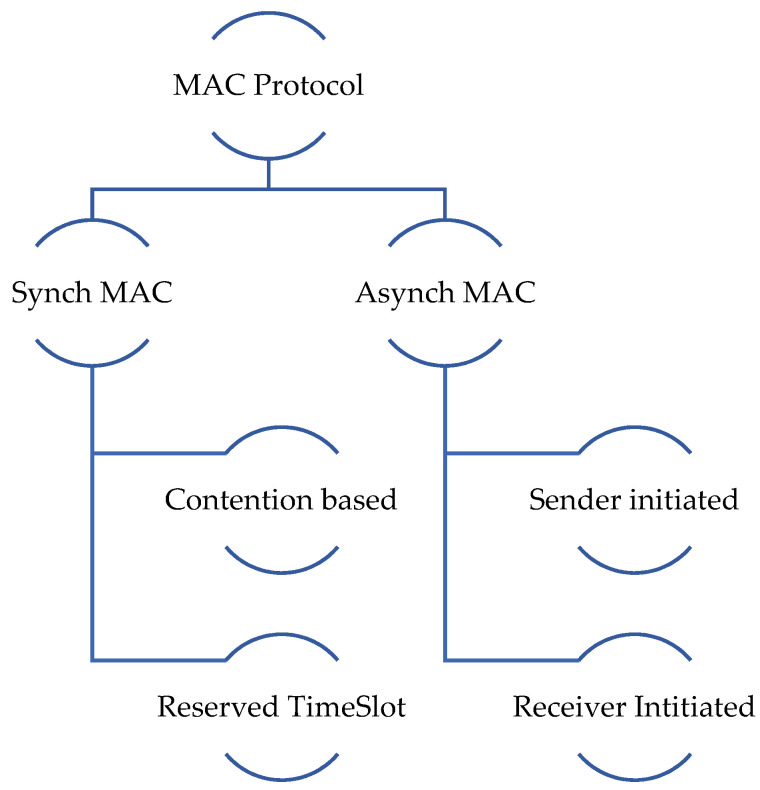
Classification of MAC protocols.

**Figure 3 sensors-22-02129-f003:**
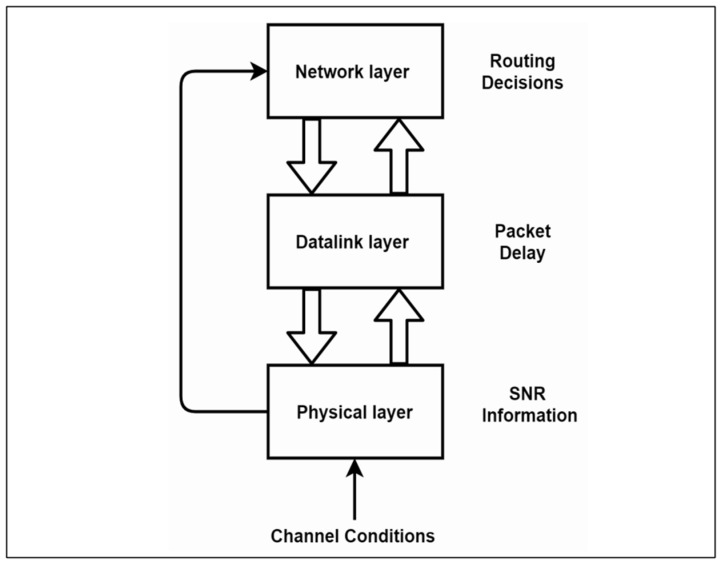
Conceptual design of cross-layer network.

**Table 1 sensors-22-02129-t001:** General approaches for the MAC protocol.

Reference Paper	Approach	Type	Contribution	Results
[29]	Sensor-MAC (S-MAC)	Synch-TMDA	Propose the periodic sleeping of nodes in channel signaling message passing	Reduces contention latency and saves energy
[30]	S-MAC protocol	Asynch-CSMA	Converting its duty cycle to adaptive based on the non-occurrence of activation for a time threshold TA	Describes the limitations of existing S-MAC
[31]	ADMC-MAC	Synch-TMDA	Improves data transmission performance based on traffic conditions by taking into account the size of the node queue and improving the energy-efficient performance	ADMC-MAC enhances the residual energy saving
[32]	Timeout-MAC (T-MAC)	Asynch-CSMA	Enables adaptive duty cycle active/sleep based on listening for time period TA and sleep if no event has occurred	Reduces energy and minimizes collisions
[33]	Hybrid MAC Protocol using a Scheduling based dynamic Sleeping	Synch-TMDA	Enabling the scheduling table of nodes’ sleep/wake up time and dividing the channel into a set of TDMA slots where some slots are provided for the contention of sub-nodes using CSMA/CA	Improves the network throughput and enhances energy conservation
[34]	Zero collision MAC (ZC-MAC)	Asynch-CSMA	Zero collisions based on a medium decomposition to a pre-defined number of slots of the same size as the number of nodes considering the slots that collided in the previous cycles	ZC outperforms both CSMA and TDMA at high and low loads
[35]	Designated learning-MAC	Synch-TMDA	Learning the probability of selecting slots for transmission based on success and collision	Reduces the number of collisions
[36]	Modified CSMA/CA	Asynch-CSMA	Enabling deterministic back-off after successful transmission	Reduces the number of collisions
[37]	TSCH MAC	Synch	Presenting a centralized mechanism to schedule TSCH time-slots with the optimal usage of resources	Outperforms previous analogous systems in terms of performance
[38]	DSME-GTSs	Synch	Beacon slot collision problem based on proposing a non-conflicting beacon scheduling mechanism using association order	Minimizes the number of time-slots used while maximizing the usage of available channels

**Table 2 sensors-22-02129-t002:** Game theory-based MAC approaches.

Reference Paper	Approach	Type	Contribution	Results
[39]	Simplified game-theoretic MAC (G-MAC)	Asynch-CSMA	Tuning the contention |window based on game theory model assigned to each node	The throughput of the system increases, the delay and packet-loss-rate are reduced, while maintaining relatively low energy consumption
[40]	Game theory based ETDMA	Synch-TMDA	Game-based energy-efficient TDMA (G-ETDMA) for intra-cluster WSN	Reduces the energy consumption to maximize the lifetime of the sensor network
[41]	Game theory distributed approach	Asynch	Optimizes the sleep interval between consecutive wake-ups by reducing idle- listening time through a dynamic duty-cycling technique	Minimizes the energy waste of the most power-hungry nodes of the network
[42]	Game theory framework	Asynch-CSMA	Proposes a generalized optimization framework to map the cost of each player onto protocol-specific MAC parameters.	Achieves a fair energy-delay performance trade-off under the application requirements
[43]	Multilayer DS-MAC using game theory optimization approach	Asynch	The use of multilayer nodes with distributed MAC (DS-MAC) where the listening time of the nodes is controlled based on neighboring communication. Route loss constraints are addressed using game theory optimization while selecting routes towards base stations (BS)	Improves the energy consumption, throughput, and network lifetime

**Table 3 sensors-22-02129-t003:** Heuristics-based MAC approaches.

Reference Paper	Approach	Type	Contribution	Results
[44]	Sensor deployment using optimization techniques and scheduling approach	Synch-TMDA	A heuristic scheduling algorithm based on grouping the sensors and considering the residual energy in each sensor	Increases the network lifetime
[45]	Energy-efficient scheduling for data aggregation and transmission for network node using time division multiple access (TDMA)	Asynch-CSMA	Multi-channel scheduling algorithm formulated the ILP optimization of integer linearly to find the minimum limit for the network energy consumption	Reduces the magnitude of the computation time
[46]	It considers the problem of activating links in a rechargeable Wireless Sensor Network (rWSN)	Asynch-CSMA	Heuristic-based scheduling using a greedy approach was proposed considering the aspect of energy harvesting using a greedy approach	Reduces the number of charge/ discharge cycles and increases link schedules
[47]	Optimum ConvergeCast Scheduling in WSN	Asynch-CSMA	A mathematical model for solving the converge cast problem in WSN using pricing problems in a round robin fashion due to the NP-hard nature of the optimization	Provides better schedules for nodes
[48]	WSN scheduling depending on the network virtualization concept is proposed using a greedy approach	Asynch-CSMA	Classifies the networks into various profiles; a single profile indicates nodes group sharing the same channel requirements or characteristics. Then, an optimization process is implemented to increase profile utilization	Increase network throughput

**Table 4 sensors-22-02129-t004:** Meta-heuristic-based MAC approaches.

Reference Paper	Approach	Type	Contribution	Results
[49]	An online optimization framework of CSMA/CA for dealing with shared time-slot of ISA100.11a has been implemented	Asynch-CSMA	The design of the solution includes two types of information: the first one is the number of nodes in each group and the second one is the packet lifetime for each group’s objective function, which includes the QoS	Enhances PDR and end-to-end delay
[50]	Applying metaheuristic approaches to solve the scheduling problem	Asynch	The solution is designed to encode which channel and time-slot is allocated for a certain node. The objective function is formulated as the E2E delay	Reduces the end-to-end delay
[51]	A Fuzzy Logic Approach by using Particle Swarm Optimization for Effective Energy Management in WSN	Asynch	Uses PSO for optimizing the membership functions of fuzzy system that is responsible for minimizing the energy consumption in IEEE 802.15.4	Improves network throughput and end-to-end delay
[52]	Evolutionary Algorithm for Scheduling in WSN	Asynch-CSMA	Particle swarm optimization was integrated with the evolution algorithm for the goal of minimizing the delay	Decreases the fitness value, which can minimize end-to-end delay

**Table 5 sensors-22-02129-t005:** Machine learning-based approaches.

Reference Paper	Approach	Type	Contribution	Results
[53]	A recurrent neural network mac protocol	Asynch-CSMA	Makiuchi is a cognitive MAC protocol that uses a recurrent neural network to represent channels occupied and determine the precise moment of transmission opportunity. The data used for the training process came from a real-world experiment that used Software-Defined Radio (SDR) to monitor a Wi-Fi channel	Enhances network throughput and decreases the number of collisions
[54]	Realtime scheduling and power allocation using deep neural networks	Asynch	An allocation was carried out for the channel resources among multiple small cells in order to enable users to schedule an uplink or downlink for each cell at a time-slot. A deep Q-network (DQN) estimates a suitable schedule; then, a DNN allocates power to the corresponding schedule	Decreases end-to-end delay and packet loss
[55]	A neural-network-based MF-TDMA MAC scheduler	Synch-TMDA	Frequency and time-slot resources are allocated for sensors based on an NN trained to predict the best channel–slot pair for transmission	Decreases the number of collisions
[56]	Cognitive radio and machine learning for intelligent wireless communications	Asynch	Spectrum sensing and access of supervised, unsupervised, and reinforcement learning for wireless communication	Enhances spectrum efficiency and energy efficiency
[57]	Hybrid MAC Protocol data collection in WSN	Asynch-CSMA	In the UAV-based wireless network, a hybrid medium access protocol (MAC) is used to collect data. There are two important times in the frame: traction and gathering. CSMA/CA is employed throughout the registration procedure, and the likely notice schedule is assigned to each node recorded during the collection period	Increases network throughput and data packet delivery

**Table 6 sensors-22-02129-t006:** Machine learning-based approaches.

Reference Paper	Approach	Type	Contribution	Results
[59]	A cross-layer design for a self-healing, multihop, and self-formation network	Synch-TMDA	Provides a multi-layer design for a for a self- healing, multihop, and self-formation network to minimize the time of the call setup. This implements collision-free communication and reuses the empty slots of the Time Division Multiple Access (TDMA) protocol, which results in a low throughput and high delay	Increases network throughput and decreases the number of collisions
[60]	A cross-layer design of physical MAC using power adaptation transmit and antenna selection	Async-CSMA	Proposes a cross-layer MAC design using power adaptation transmit and antenna selection. This is implemented depending on the transmit power adaptation and antenna selection	Improves the network throughput
[61]	Priority-based multiple access (PBMA)	Synch-TMDA	A novel full-duplex MAC protocol called Priority-Based Multiple Access (PBMA) based on priority messaging between different nodes. The PHY layer, analyzes incomplete full-duplex (FD) simultaneous transmissions and scans and mathematically formulates dynamic thresholds to determine the channel status before and during transmissions	Increases the network throughput
[62]	Cross-layer design for optimizing transmission reliability and energy consumption	Synch	Three types of developments were made: the first one is optimizing the transmission power of the nodes, the second one is selecting relay nodes, and the third one is synthesizing the cross-layer method	Enhances transmission reliability and maximizes network lifetime
[63]	FAMACROW: a hybrid fuzzy and ant colony optimization	Async	Proposes a Fuzzy and Ant Colony Optimization, which mainly depends on an Unequal Clustering Cross-Layer Protocol and Combined MAC and Routing algorithms for Wireless Sensor Networks (FAMACROW), which include various nodes that send collected data to a Master Station. The selection of the cluster head, clustering process, and internal cluster routing protocols is incorporated using FAMACROW	Increases network throughput and maximizes network lifetime

## Data Availability

The data that support the findings of this study are available from the corresponding author, R.H., upon reasonable request.

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
