# Peer review of "Conceptual Framework for Future WSN-MAC Protocol to Achieve Energy Consumption Enhancement"

_sensors, 2022, doi:10.3390/s22062129_

Round 1

Reviewer 1 Report

This paper reviews the recent approaches to enhance the performance of MAC protocol in WSN. The paper well investigates related papers. However, the reviewer thinks that some revisions are needed for publication from the viewpoint of the organization as a review paper. So, please answer the following comments: 

Mandate revisions: 

  1. In Section 2, the author claims that this paper investigates state-of-the-art approaches proposed to enhance the performance of MAC protocol in WSN. However, the above explanation is so ambiguous that it is difficult for readers to understand the contribution from the past review papers, such as [15-18]. So, please add more detailed explanations related to state-of-the-art approaches to enhance the performance into Section 2.  
  2. In sections 4.2, 4.3, 4.3, and 4.5, the author introduces just the contents of each related paper. The author does not refer to the characteristics, such as merit and demerit, of each approach in WSN. So please add explanations of the characteristics of each approach.  
  3. The title of section 4 "Optimization of MAC protocol" sounds strange from the viewpoint of the full organization of this paper and the connection with section 3. If the motivation of this paper is to introduce state-of-the-art approaches to enhance the performance of MAC protocol in WSN, it is better to change the title to ``State-of-the-art approaches to enhance the performance of MAC protocol.'' Please reconsider the title of Section 4 and change it if necessary. Then, please refer to the difference between the past approaches with state-of-the-art approaches. If the author keeps the title, please refer to the optimization of MAC protocols in Section 2. Then, please add an explanation about the difference between the past approaches with state-of-the-art approaches.
  4. The author thinks that it is difficult for readers to comprehend the organization of section 4. In the first paragraph of section 4, please add the abstract of the organization of section 4. 

Minor revisions: 

  1. If the author claims state-of-the-art approaches, it is better to include the publication year of each paper into each table. However, the space is limited. So, please reconsider the presentation of each Table.  
  2. On page4, Section 4, the end of the first paragraph, threshold 'TA'. The reviewer thinks that there is no explanation of abbreviations of 'TA'. So, please add the explanation of abbreviations of 'TA' if there is not.  
  3. It seems there remain some abbreviations without detail. So, please check carefully. 

Reviewer 2 Report

Strong point: This paper presents a state-of-the-art review of recently proposed WSN MAC protocols. The study included different methods and approaches proposed to enhance main performance factors. They provided a comparison table used to list further details about using approaches and algorithms to enhance different performance issues, such as network throughput, end to end delay, and packet drop, which can be translated into energy consumption.

But I have some concerns that can be properly addressed before this paper can be considered for publication.

  1. The abstract should state briefly the purpose of the research, the principal results and major conclusions. The abstract needs improvement in terms of clearly highlighting the contribution.
  2. In Key words, I would add IoT, Network performance.
  3. The authors have missed to highlight their contribution in the last part of the introduction section;
  4. The contribution section should be removed and the descriptions of the rest of the paper should be added to the end of the introduction section.
  5. Section 4 says, “The literature has concentrated on developing optimization of MAC protocols in the wireless sensors ‘networks for critical applications such as emergency alerting and monitoring missions as well as health care mission.” In my opinion some more application-oriented studies in WSN should be covered. Some relevant papers like given below can be read and cited in the paper:
    • Centralized Graph based TSCH Scheduling for IoT Network Applications N Choudhury, MM Nasralla, P Gupta, IU Rehman 2021 IEEE Intl Conf on Parallel & Distributed Processing with Applications …
    • A comprehensive overview of AI-enabled music classification and its influence in games, T Yang, S Nazir
    • A Proposed Resource-Aware Time-Constrained Scheduling Mechanism for DSME based IoV Networks N Choudhury, MM Nasralla 2021 IEEE 94th Vehicular Technology Conference (VTC2021-Fall), 1-7
    • Internet of Things Based Intelligent Techniques in Workable Computing: An Overview, J Guo, S Nazir, Scientific Programming 2021
    • A Beacon and GTS Scheduling Scheme for IEEE 802.15. 4 DSME Networks N Choudhury, R Matam, M Mukherjee, J Lloret IEEE Internet of Things Journal

The title is a bit generic and vague, I would consider revising;

6. Kindly ensure the abbreviations are all defined, for example, CLD.

7. The paper requires intensive grammar check. There are many grammatical and typo mistakes. The article should be carefully revised For example in line 72, “This paper investigates stat of art….” It should be “state…..”. and Line 383 “This paper mainly address” it should be “addresses”…

8. Figures are blurred, they should be redesigned, also, please add a taxonomy figure that shows the state of the art scheduling rules in section 4.

9. I would suggest to add one section called conclusion and future work rather than section 6. Also limitation should be discussed in the research gap.

Reviewer 3 Report

Dear Authors

Please find attached my comments.

Round 2

Reviewer 1 Report

I agree the authors' revisions. 

Reviewer 2 Report

Thank you.

Reviewer 3 Report

Thanks authors for addressing my comments.

I was convinced with the updated version!

Good luck!